# Canaloplasty in Pigmentary Glaucoma: Long-Term Outcomes and Proposal of a New Hypothesis on Its Intraocular Pressure Lowering Mechanism

**DOI:** 10.3390/jcm9124024

**Published:** 2020-12-12

**Authors:** Paolo Brusini, Veronica Papa

**Affiliations:** Department of Ophthalmology, “Città di Udine” Health Clinic, Viale Venezia, 410, 33100 Udine, Italy; veronica.papa@policlinicoudine.it

**Keywords:** pigmentary glaucoma, canaloplasty, trabecular meshwork

## Abstract

This study presents the long-term results on canaloplasty in a group of patients affected by pigmentary glaucoma, and studies the progression of the disease after surgery. Material and methods: Twenty-nine eyes of 25 patients with pigmentary glaucoma in maximum tolerated medical therapy with significant visual field damage progression underwent canaloplasty and were followed up to 11 years (mean 59.8 ± 30.1 months). All patients underwent a complete ophthalmic examination every 6 months. Results: The pre-operative mean intraocular pressure (IOP) was 31.8 mmHg ± 10.9 (range 21–70) with an average of 3.3 medications. After 1, 2, 3, and 4 years, the mean IOP was 15.9 ± 4.0, 14.4 ± 7.3, 14.1 ± 2.1, and 15.7 mmHg, respectively, with 0.4, 0.5, and 0.7 medications, respectively. Four patients underwent trabeculectomy after 3 to 30 months due to uncontrolled IOP. Gonioscopy showed a significant reduction of pigment in trabecular meshwork in all cases, starting from the sixth month. In some cases, the pigment was almost completely reabsorbed after two years, suggesting an accelerated transit and escape of the granules through the trabecular spaces. Conclusions: Canaloplasty seems to be a reasonable option in treating patients affected by progressive pigmentary glaucoma. The reabsorption of pigment granules from the trabecular meshwork could, at least in part, explain the relatively high success rate observed after this surgical procedure.

## 1. Introduction

Pigmentary glaucoma (PG) is a quite uncommon type of secondary glaucoma due to the dispersion and accumulation of pigment granules coming from the posterior iris surface onto the trabecular meshwork [1,2]. The concavity of the mid-peripheral iris causes a reverse pupillary block, which leads to irido-zonular contact with pigment dispersion [3,4]. PG typically affects young males with moderate myopia, and is characterized by high diurnal intraocular pressure (IOP) fluctuations, and possible sudden rises of IOP with subjective symptoms, such as iridescent halos, headache, and blurred vision.

It is known that trabecular pigmentation can decrease over time (“burn-out phase”) due to the macrophage activity responsible for the trabecular meshwork clearance of pigment and debris, leading to improved glaucoma control and, in rare cases, even a complete remission of the disease [5,6,7,8].

In some cases, an active dispersion of clinically detectable pigment will continue to occur over time, thus rendering periodic optic nerve and visual function evaluations necessary, in addition to medical therapy in the presence of pathology [9,10,11,12]. Surgical treatment may be needed to avoid structural and functional damage progression when non-invasive therapy proves to be insufficient. Several surgical techniques can be used, including trabeculectomy and various kinds of non-perforating procedures [13,14,15,16,17,18].

Canaloplasty is a surgical procedure designed to restore the eye’s natural outflow drainage pathways. Surgery involves the use of a microcatheter to inject high-viscosity sodium hyaluronate in Schlemm’s canal. A 10-0 Prolene suture is left within the canal to ensure an effective and long-lasting 360° dilation effect, which causes an increase of aqueous humor outflow. The mid- and long-term outcomes of this technique in different types of glaucoma have been reported in numerous published studies [19,20,21,22,23]. Despite lower efficacy in reducing IOP than trabeculectomy, the complication rate of canaloplasty is significantly lower [24,25,26,27]. Based on the fact that pigmentary glaucoma affects patients often with mild to moderate visual field damage, with no need of a particularly low target IOP, in addition to the risk of a filtering procedure in young patients, canaloplasty can be a reasonable choice in this type of glaucoma. According to our experience, the long-term results seem to be satisfactory in a high percentage of cases. Based on our clinical observations in these surgical cases, we hypothesize that canaloplasty could also interfere with pigment deposition on trabecular meshwork and its reabsorption. Considering that canaloplasty works by enlarging the Schlemm’s canal and the aqueous outflow pathways [28], we speculate that it could also have a role in speeding up pigment reabsorption. The purpose of this study was to verify the long-term outcomes of canaloplasty and to assess its ability to accelerate pigment removal from trabecular tissue of patients with pigmentary glaucoma, thus providing an etiological treatment.

## 2. Study Design

This is a prospective, observational, monocentric, non-randomized study conducted on a group of patients affected by pigmentary glaucoma.

## 3. Material and Methods

Twenty-nine eyes of 25 patients (22 males and 3 females; mean age 52.6 ± 9.6 years; range 32–65 years) affected by pigmentary glaucoma with a not well-controlled IOP (despite maximum tolerated medical therapy with significant visual field damage progression) documented in at least two consecutive visual field tests, were serially enrolled in the study. A low-power, half-circumference selective laser trabeculoplasty (SLT) was performed in five patients 3 to 5 months before; however, results after treatment were not satisfactory. All patients underwent canaloplasty between February 2009 and January 2019 in two different centers (S. Maria della Misericordia Civil Hospital of Udine and “Città di Udine” Health Clinic, Udine, Italy). All operations were performed by the same surgeon (P.B.).

The study was in compliance with the tenets of the Helsinki’s Declaration, and written informed consent was obtained from all participants prior to testing. The study was in compliance with institutional review boards (IRBs) and HIPAA requirements of the Azienda Ospedaliero-Universitaria “S. Maria della Misericordia”, Udine, Italy. We certify that all applicable institutional and governmental regulations concerning the ethical use of human volunteers and patients were followed during this research.

Inclusion criteria included patients affected with pigmentary glaucoma not adequately controlled with maximum tolerated medical therapy, with early to moderate functional loss (Glaucoma Staging System 2, stage 1 to 3) [29] and a significant progression of the visual field defect in two consecutive tests analyzed with the Glaucoma Progression Analysis 2 program (Carl Zeiss Meditec Inc., Dublin, CA, USA).

Exclusion criteria included patients with previous ocular surgery, other major eye diseases, narrow angle eyes, and unwillingness to undergo this procedure. All patients were older than 18 years.

Canaloplasty was performed under local anesthesia, as previously described [21,22]. In brief, surgery starts with a fornix-based conjunctival flap and a 3 × 4 mm superficial scleral flap, which is dissected forward into the clear cornea for 1.5 mm. A deep scleral flap is then created, which opens into Schlemm’s canal. This flap is removed and a microcatheter (iTrack™, Ellex iScience, Inc., Freemont, CA, USA), connected to a flickering red laser source for easy identification of the distal tip through the sclera, is then inserted and pushed forward within Schlemm’s canal for the entire 360° until it comes out of the other end of the canal opening. A double 10-0 Prolene suture is then tied to the distal tip, and the microcatheter is withdrawn and pulled back through the canal in the opposite direction. A screw-driven syringe is used to deliver a small amount of high–molecular weight hyaluronic acid in the Schlemm’s canal while the catheter is withdrawn. The sutures are then knotted under tension to further inwardly distend the trabecular meshwork. The superficial scleral flap is tightly sutured with five to seven 10-0 polyglactin 910 stitches to ensure a watertight closure, and the conjunctival flap is then sutured to complete the surgery.

The follow-up included visits at day one, and after 1 week, 1 month, 3 months, and thereafter, every 6 months. The control visits included IOP measurement with a Goldmann applanation tonometer, visual acuity measurement, and fundus examination. Starting from the third month, patients underwent gonioscopy with photography, visual field examination (Zeiss–Humphrey 24-2 SITA Standard test), and optic nerve and retinal nerve fiber layer assessment using a Spectral Domain Optical Coherent Tomograph.

## 4. Results


In three eyes, the procedure was converted into a viscocanalostomy (viscodilation of Schlemm’s canal without suture positioning), due to the impossibility of completing the 360° cannulation. These three eyes were excluded from statistical analysis. Four patients underwent a trabeculectomy after 3 to 30 months due to an uncontrolled IOP. The last IOP value before trabeculectomy was considered for statistical analysis.The follow-up ranged between 19 and 137 months (mean 59.8 ± 30.1), excluding the four eyes that underwent trabeculectomy).The pre-operative mean IOP was 31.8 mmHg ± 10.9 (range 21–70) with an average of 3.3 medications. The IOP values along various follow-up periods are shown in Figure 1.



After 1, 2, 3, and 4 years, the mean IOP was 15.9 ± 4.0, 14.4 ± 7.3, 14.1 ± 2.1, and 15.7 ± 6.8 mmHg, respectively, with 0.4, 0.5, 0.6, and 1.1 medications, respectively (Figure 2).


The preoperative and three-year postoperative IOP values are plotted in Figure 3.

The Kaplan–Meier survival curve up to four years is shown in Figure 4. 


The complete and qualified (in brackets) success rate—defined as an IOP ≤ 21 mmHg—after 1, 2, 3, and 4 years was 80.0% (96.0), 57.9% (84.2), 62.3% (92.9), and 50% (87.5), respectively. Considering an IOP ≤ 18 mmHg as the criterion for success, the percent of success in our surgical patients over 4 years were: 68.0% (84.0), 57.9% (84.2), 57.1% (85.7), and 50% (87.7), respectively. Visual field damage showed no significant progression in all patients during the follow-up period.Gonioscopy showed, in all cases, a significant reduction of pigment dispersion in the trabecular meshwork starting from the sixth month in all cases (Figure 5A–D and Figure 6A–D).


In some cases, the pigment was almost completely reabsorbed after one year, suggesting an accelerated transit and escape of the granules through trabecular spaces. However, we could not demonstrate a clear relationship between the reduction of trabecular pigmentation and IOP lowering. 

## 5. Discussion

Pigmentary glaucoma management and treatment can sometimes be challenging, due to the particular and sometimes capricious behavior of this disease. In most cases, medical therapy is able to adequately control IOP; but sometimes, other, more aggressive options are required to avoid progressive worsening of visual field damage. Argon Laser Trabeculoplasty and Selective Laser Trabeculoplasty may prove to be effective in some patients [30,31,32], but the risk of very high IOP peaks within a few hours after the treatment—at times untreatable with the need of an urgent surgery—must be taken into account. Surgical treatment should be considered when other options have failed. In these cases, traditional trabeculectomy is usually the preferred choice of surgery [10,13], even if several early and late complications, mostly bleb-related, can occur. Other possible surgical procedures include trabecular aspiration [14], deep sclerectomy, ab interno trabeculotomy [15,16], implant of minimally invasive glaucoma surgery (MIGS) devices [17,18], and canaloplasty. The latter technique seems to be a very interesting option considering its safety profile, especially in this particular type of patients. The lack of a subconjunctival filtering bleb, and the physiological mechanism in which the intervention works are the main advantages of this procedure, which are widely reported in the literature. The mid- and long-term outcomes of canaloplasty in pigmentary glaucoma seem to be satisfactory, even if very few studies specifically concerning this topic have been published to date [33]. Our results confirm that canaloplasty is a safe and effective surgical procedure which should be taken into consideration as a possible option in patients with pigmentary glaucoma that are not responsive to medical therapy, even if a relatively high percentage of patients still needs medical treatment after several years (>30% after 3 years). The accidental discovery that the pigmented granules present in the trabecular meshwork progressively decrease until they disappear almost completely a few months after a canaloplasty sheds new light on the mechanisms that lead not only to the control of IOP, but also, at least in a fair percentage of eyes, to a complete recovery of this secondary glaucoma, as spontaneously happens in few patients after many years. Is this a definitive healing? Only long-term prospective studies on a larger sample of patients and a proper analysis of histological findings (which are not very easy to perform) can, of course, address this question. It should be remembered, however, that, according to some studies [34,35], the development of the chronic glaucomatous condition cannot be directly and entirely attributed to pigment accumulation in the juxtacanalicular tissue; thus, the disappearance of pigment does not automatically mean that the aqueous humor pathways are well-functioning. Loss of trabecular cells, fusion of trabecular lamellae with collapse of intertrabecular spaces, increase in extracellular material, and obliteration of the canal are all factors that can potentially contribute to the development of ocular hypertension regardless of trabecular pigmentation, especially if the disease lasts for a long time [36]. These considerations can explain why in some eyes, the IOP remains high after canaloplasty even if the trabecular meshwork appears to be free of pigment. On the other hand, it is reasonable to speculate that early surgical treatment (which frees the trabecular meshwork from the pigment) before chronic and irreversible damage occurs in this structure, could be beneficial in this type of glaucoma.

This study has several limitations. It is a non-randomized study with no control group operated with another surgical technique. For this reason, it is not possible to compare the results obtained with canaloplasty with other surgeries, nor affirm that only canaloplasty leads to a reabsorption of trabecular pigment. It is important to note, however, that in some patients in which one eye underwent trabeculectomy, and the second canaloplasty, a reduction in trabecular pigmentation was observed postoperatively only in the eye that underwent canaloplasty. In another patient with pigmentary glaucoma operated with a canaloplasty in the right eye two months after a deep sclerectomy in the left, a significant reduction in pigmentation was observed three years after the operation in only the right eye (unpublished personal observations).

Another limitation relates to the relatively small number of patients that were considered in the study. Considering that this type of disease is not very common, we believe that our sample can offer information that could be clinically useful. 

Moreover, the follow-up, even if it reaches up to 5 years on average, may be not sufficient to draw conclusive considerations. 

In addition, the lack of histological findings does not assist in providing possible mechanisms behind the apparent washing of trabecular meshwork. 

## 6. Conclusions

Canaloplasty seems to be a reasonable option to treat patients affected by progressive pigmentary glaucoma who do not respond to medical treatment. The progressive reabsorption of pigment granules from the trabecular meshwork can, at least in part, explain the quite high success rate observed after this surgical procedure.

## Figures and Tables

**Figure 1 jcm-09-04024-f001:**
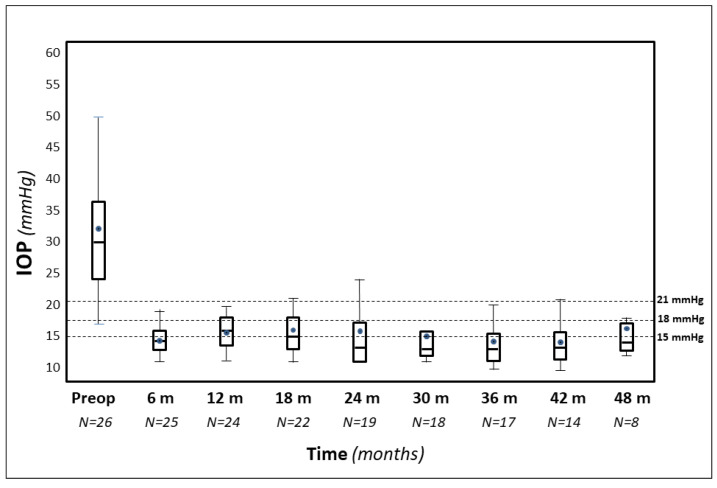
Box-plot representation of intraocular pressure (IOP) values over four years of follow-up.

**Figure 2 jcm-09-04024-f002:**
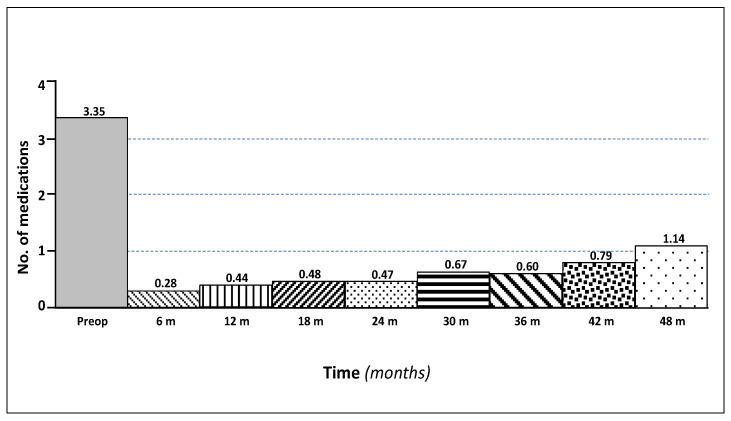
Number of medications at baseline and over a four-year follow-up.

**Figure 3 jcm-09-04024-f003:**
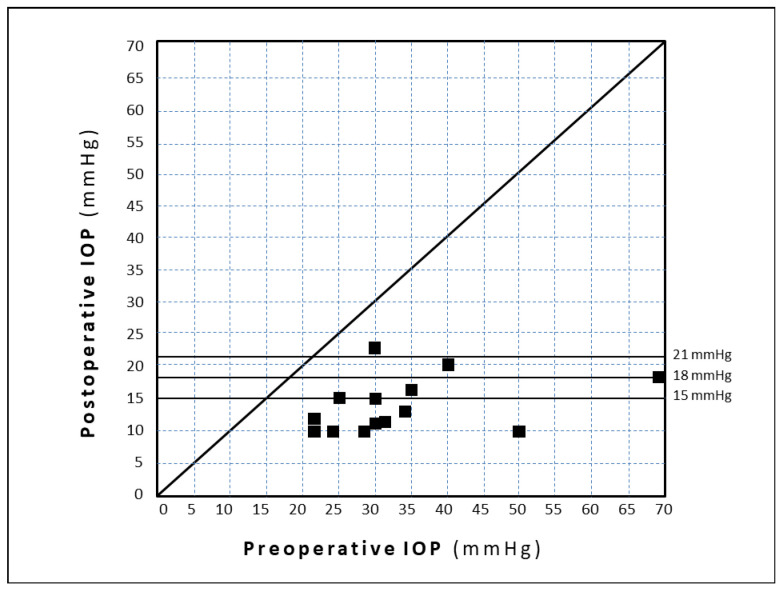
Scattergram showing the IOP values after three years vs. pre-operative IOP values.

**Figure 4 jcm-09-04024-f004:**
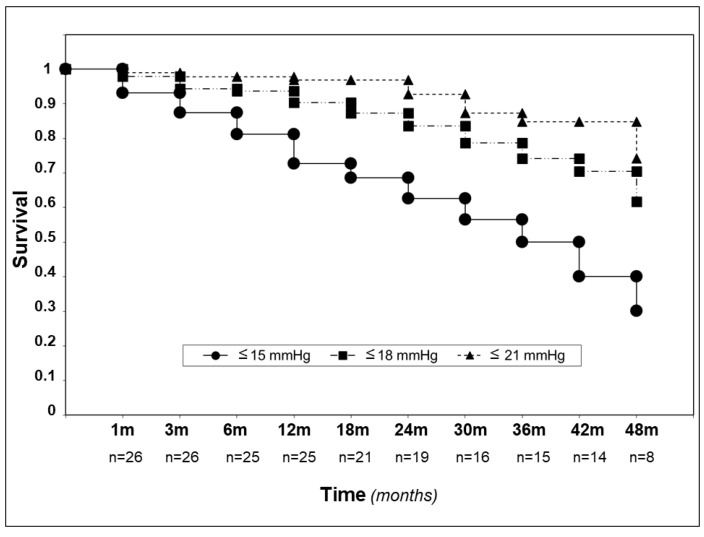
Kaplan–Meier graph according to three different endpoints (IOP ≤ 21, 18, and 15 mm/Hg).

**Figure 5 jcm-09-04024-f005:**
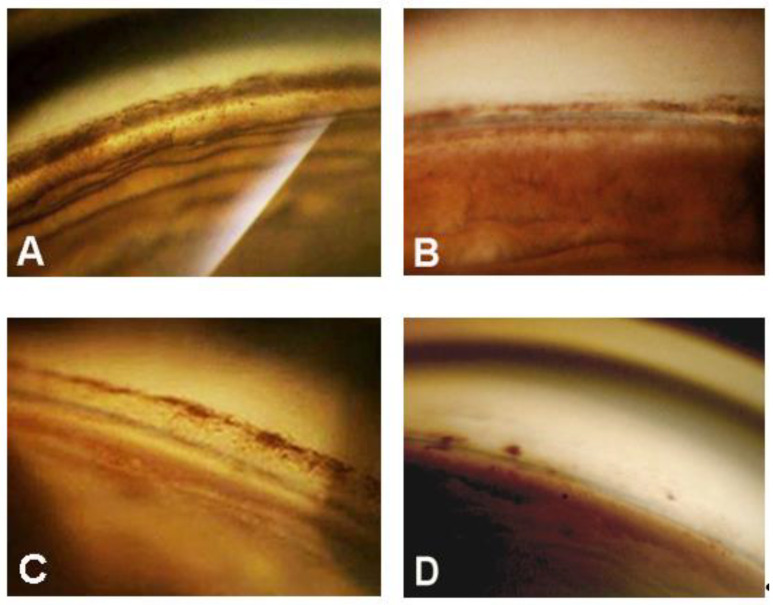
(**A**) Preoperative gonioscopic view of the trabecular meshwork in a 46-year-old man with pigmentary glaucoma; B and C significant reduction of trabecular pigment, one year (**B**) and two years (**C**) after canaloplasty; (**D**) almost complete reabsorption of trabecular pigment three years after surgery.

**Figure 6 jcm-09-04024-f006:**
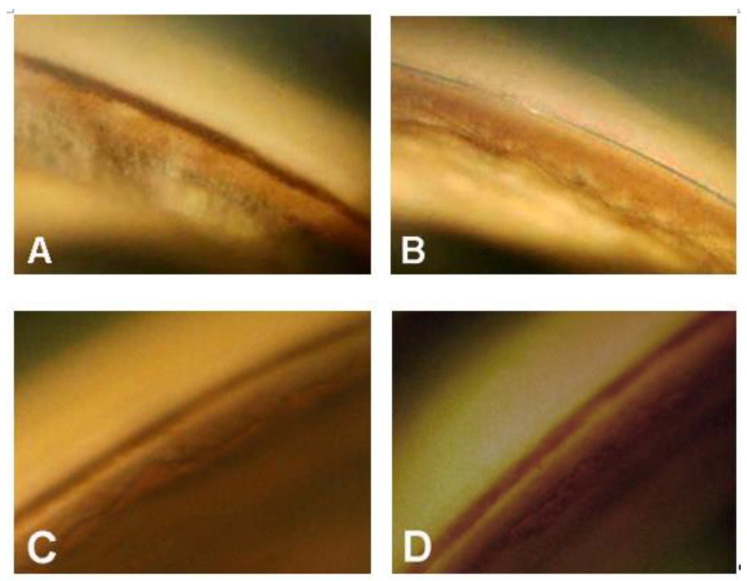
(**A**) Preoperative gonioscopy in a 62-year-old woman with early pigmentary glaucoma (right eye); (**B**) same eye one year after canaloplasty—no pigment is present in the trabecular meshwork; (**C**) left eye of the same patient at the baseline; (**D**) gonioscopy of the left eye after two years—no modification in pigment distribution is present.

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
