# Peer review of "Canaloplasty in Pigmentary Glaucoma: Long-Term Outcomes and Proposal of a New Hypothesis on Its Intraocular Pressure Lowering Mechanism"

_jcm, 2020, doi:10.3390/jcm9124024_

Round 1

Reviewer 1 Report

This referee has no major criticisms to add to those already made by the Authors in their discussion. Accordingly, there is limited novelty in the study that has been designed well but suffers of very limited power. In addition, per se the technique has been largely validated with superimposable success rate (see summarized data in NICE website). However, with all the caveats above, the long term follow up of the outcomes set is what it is indeed needed. In this respect, this research effort can be considered as a trigger.

It looks as the Authorship is different with respect to the summarized by the editorial office. Please, check. 

Author Response

This referee has no major criticisms to add to those already made by the Authors in their discussion. Accordingly, there is limited novelty in the study that has been designed well but suffers of very limited power. In addition, per se the technique has been largely validated with superimposable success rate (see summarized data in NICE website). However, with all the caveats above, the long term follow up of the outcomes set is what it is indeed needed. In this respect, this research effort can be considered as a trigger.

It looks as the Authorship is different with respect to the summarized by the editorial office. Please, check. 

We thank the Reviewer for the favorable comments. I changed affiliations in 2016. We checked the Authorship and confirm that it is correct.

Reviewer 2 Report

The proposed article presents some typing errors such us the repetitions of words in lines 53, 71 and 181. In my opinion this is an interesting article despite the limitation already described by the author.

Author Response

The proposed article presents some typing errors such us the repetitions of words in lines 53, 71 and 181. In my opinion this is an interesting article despite the limitation already described by the author.

Many thanks for the positive comments by the Reviewer. We apologize for the typing errors. The manuscript has been checked and corrected. The manuscript was thoroughly checked and corrected by a native English speaker

Reviewer 3 Report

The authors present a report on the prospective outcomes of suture canaloplasty in patients with pigmentary glaucoma with IOP elevation refractory to medical therapy. The study title should be edited to better reflect what the study evaluated; the study was not powered or designed to evaluate  the question can canaloplasty definitely “heal” pigmentary glaucoma?" The study evaluated the success of canaloplasty in Medically Refractory Pigmentary Glaucoma. The study authors acknowledge the main limitations of the study as the small sample size, single-center and variably follow-up period. The authors should also discuss in more detail the lack of consideration of a partial-fluence SLT effect prior to surgical intervention and no commentary on how patient selection was completed to avoid selection bias (i.e. serial patient enrollment). 

The authors should add the manufacturer information for the of the illuminated microcatheter probe used in their surgical technique. 

The report would also require moderate English editing as there are several sentences with poor grammar and duplicate phrases. The reports will require professional English editing. 

Author Response

The authors present a report on the prospective outcomes of suture canaloplasty in patients with pigmentary glaucoma with IOP elevation refractory to medical therapy. The study title should be edited to better reflect what the study evaluated; the study was not powered or designed to evaluate  the question can canaloplasty definitely “heal” pigmentary glaucoma?" The study evaluated the success of canaloplasty in Medically Refractory Pigmentary Glaucoma. The study authors acknowledge the main limitations of the study as the small sample size, single-center and variably follow-up period. The authors should also discuss in more detail the lack of consideration of a partial-fluence SLT effect prior to surgical intervention and no commentary on how patient selection was completed to avoid selection bias (i.e. serial patient enrollment). 

The authors should add the manufacturer information for the of the illuminated microcatheter probe used in their surgical technique. 

The report would also require moderate English editing as there are several sentences with poor grammar and duplicate phrases. The reports will require professional English editing. 

We thank the Reviewer for the valuable comments. The title of our manuscript was meant to be somewhat provocative, but we agree with the reviewer that the study was specifically designed to evaluate the long term results of canaloplasty in this type of glaucoma. We proposed this new title: “CANALOPLASTY IN PIGMENTARY GLAUCOMA. Long-term outcomes and proposal of a new mechanism of action”. In alternative the subtitle could be “Long term outcomes and proposal of a new hypothesis on its IOP lowering mechanism”. The Editor can feel free to use either title.

With regards to SLT, the Reviewer’s comment is correct: actually, we performed a partial SLT with very low power in 5 patients. In two cases, however, we observed an acute post-laser IOP rise within three hours, which in one case requested the use of Mannitol infusion. The mid-term results of this treatment in these 5 patients were not satisfactory. Considering that the visual field loss in our patients was rapidly progressing and bearing in mind the potential risk of a laser treatment, we decided to proceed with surgery, thus skipping the SLT treatment in other cases. We added the following phrase in Material and Methods section (page 2, line 72): “A low-power half-circumference selective laser trabeculoplasty (SLT) was performed in 5 patients 3 to 5 months before, however, results after treatment were not satisfactory”.

The selection of patients for surgery was serial. The first sentence in Material and Methods section (page 2, lines 70-72) was modified as follows: “Twenty-nine eyes…with a not well controlled IOP despite maximum tolerated medical therapy were serially enrolled in the study”.

We added the manufacturer information (iTrack™, Ellex iScience, Inc., Freemont, CA, USA), as requested.

The manuscript was thoroughly checked and corrected by a native English speaker MD, PhD listed in the Acknowledgment section.